# MT-FANet: A Morphology and Topology-Based Feature Alignment Network for SAR Ship Rotation Detection

Qianqian Liu [1], Dong Li [2,*], Renjie Jiang [2], Shuang Liu [2], Hongqing Liu [3] and Suqi Li [2]

1 College of Electronic Engineering, Naval University of Engineering, Wuhan 430033, China; liuhe1984@163.com
2 School of Microelectronics and Communication Engineering, Chongqing University, Chongqing 400044, China; 202112131089t@cqu.edu.cn (R.J.); shuangliumax@cqu.edu.cn (S.L.); lisuqi@cqu.edu.cn (S.L.)
3 Chongqing Key Laboratory of Mobile Communications Technology, Chongqing University of Posts and Telecommunications, Chongqing 400065, China; hongqingliu@cqupt.edu.cn
* Correspondence: lid0705@cqu.edu.cn

**Abstract:** In recent years, ship target detection in synthetic aperture radar (SAR) images has significantly progressed due to the rapid development of deep learning (DL). However, since only the spatial feature information of ship targets is utilized, the current DL-based SAR ship detection approaches cannot achieve a satisfactory performance, especially in the case of multiscale, rotations, or complex backgrounds. To address these issues, in this paper, a novel deep-learning network for SAR ship rotation detection, called a morphology and topology-based feature alignment network, is proposed which can better exploit the morphological features and inherent topological structure information. This network consists of the following three main steps: First, deformable convolution is introduced to improve the representational ability for irregularly shaped ship targets, and subsequently, a morphology and topology feature pyramid network is developed to extract inherent topological structure information. Second, based on the aforementioned features, a rotation alignment feature head is devised for fine-grained processing as well as aligning and distinguishing the features; to enable regression prediction of rotated bounding boxes; and to adopt a parameter-sharing mechanism to improve detection efficiency. Therefore, utilizing morphological and inherent topological structural information enables a superior detection performance to be achieved. Finally, we evaluate the effectiveness of the proposed method using the rotated ship detection dataset in SAR images (RSDD-SAR). Our method outperforms other DL-based algorithms with fewer parameters. The overall average precision is 90.84% and recall is 92.21%. In inshore and offshore scenarios, our method performs well for the detection of multi-scale and rotation-varying ship targets, with its average precision reaching 66.87% and 95.72%, respectively.

**Keywords:** synthetic aperture radar (SAR); ship target detection; rotating bounding boxes; morphology features; topological structure information

## 1. Introduction

Synthetic aperture radar (SAR) technology is widely utilized in military and civilian domains owing to its all-weather capability and increasingly high-quality data advantages [1,2]. Specifically, SAR technology is gaining more interest in marine applications, such as oceanic exploration, maritime rescue, and traffic management. Despite the increasing resolution of SAR images, their manual interpretation remains a cumbersome task [3]. Recently, the developments and progress in SAR technology have led to an enhanced focus on deep-learning techniques for SAR image analysis. These techniques help individuals to better leverage SAR images for marine applications, including target detection, ship identification, and coastline monitoring [4]. As a result, SAR images can facilitate marine resource management and conservation, and can offer significant support for maritime rescue and coastal military defense warnings.

At present, SAR technology has been widely used in the field of oceanography, especially in ship monitoring and detection. Traditional SAR ship detection methods can be classified into three types: methods based on the statistical characteristics of sea clutter [5,6], which use the sea clutter information in SAR images to distinguish ship targets from sea clutter and judge the existence of ship targets based on statistical features; methods based on polarimetric decomposition [7], which use different scattering mechanisms of various objects to detect ship targets; and methods based on texture features [8,9], which use the local image features of ship targets for detection. Overall, traditional SAR ship detection methods work well for simple sea conditions within a specific range, but lack robustness and struggle with target discrimination in complex scenarios.

With the development of artificial intelligence technology, data-driven methods have become an important research direction in ship detection, and significant progress has been made in the field [10]. Therefore, in the SAR field, more and more researchers are turning to deep learning (DL)-based ship detection methods [11–14]. Regarding SAR datasets, several datasets have been published for various detection tasks [15,16]. In terms of network structure, DL-based SAR ship detection methods can be divided into anchor-based and anchor-free methods. Two-stage anchor-based detection methods mainly include region proposal extraction and bounding box classification regression, which have high accuracy, but increase the cost of detection time [17–19]. One-stage anchor-based methods have faster computation speeds and do not require region proposal extraction, but they sacrifice some accuracy [20–24]. Anchor-free detection algorithms eliminate the extra computational burden brought by anchors and have better recall [25,26]. In the field of ship detection, compared with traditional methods, deep learning-based approaches use multi-layer network structures to learn complex data representations, thus achieving higher accuracy and efficiency [15]. The superiority of this method lies in its ability to autonomously discover valuable features in the data and transform them into patterns that can be used to detect ships.

Although DL has been widely applied in SAR ship detection, some problems remain. Especially considering the diversity of ship target shapes and considerable background interference, existing object detection algorithms often struggle to achieve the ideal performance. Traditional horizontal bounding boxes (HBBs) provide unsatisfactory fitting results for oriented ships, introducing more background interference and leading to false positives or missed detections [27], as shown in Figure 1a,b. In addition, the dense arrangement of ships results in considerable overlap between HBBs, as shown in Figure 1a,c, which reduces detection accuracy.

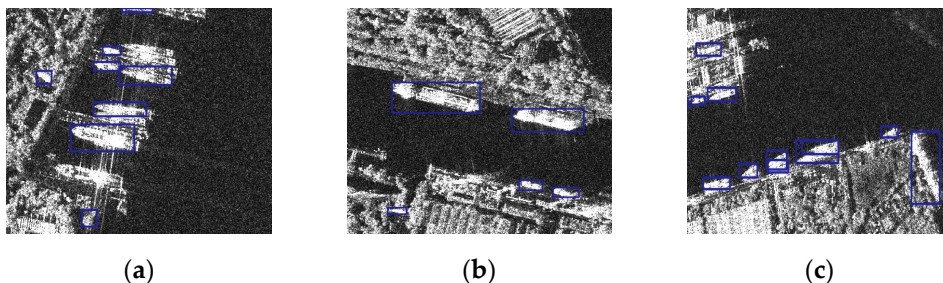

(**a**)          (**b**)          (**c**)

**Figure 1.** Horizontal bounding box detection results: (**a**) depicts the dense arrangement of ships, (**b**) shows that the HBBs contained more background noise, and (**c**) shows that the bounding boxes overlapped. There were different degrees of missed detection and false detection. The blue boxes indicate detected ship targets.

Methods based on rotated bounding boxes [28–30] can effectively suppress background interference and improve detection accuracy. However, problems such as irregular ship target morphology and insufficient utilization of inherent topological information of ship targets still need to be solved. Due to the various geometric changes (rotation, scaling, deformation, etc.) of ship targets in SAR images [29–31], as shown in Figure 2, the shapes

of ship targets exhibit a high degree of irregularity, which increases the difficulty of object detection. Further, traditional CNN cannot extract features from such irregular shapes, causing the network's to incompletely recognize the target. In addition, in previous ship target detection methods, only pixel information was usually considered. In contrast, the inherent topological information of the target was ignored, as shown in Figure 2, which also led to an insufficient understanding of the target. Therefore, when designing SAR ship target detection models, it is necessary to improve their ability to extract features from irregular target shapes and fully utilize the inherent topology information of the target to ensure that the network's feature learning is closely related to the actual target. This can result in better suppression of false positives and missed detections, thus improving the accuracy and robustness of detection.

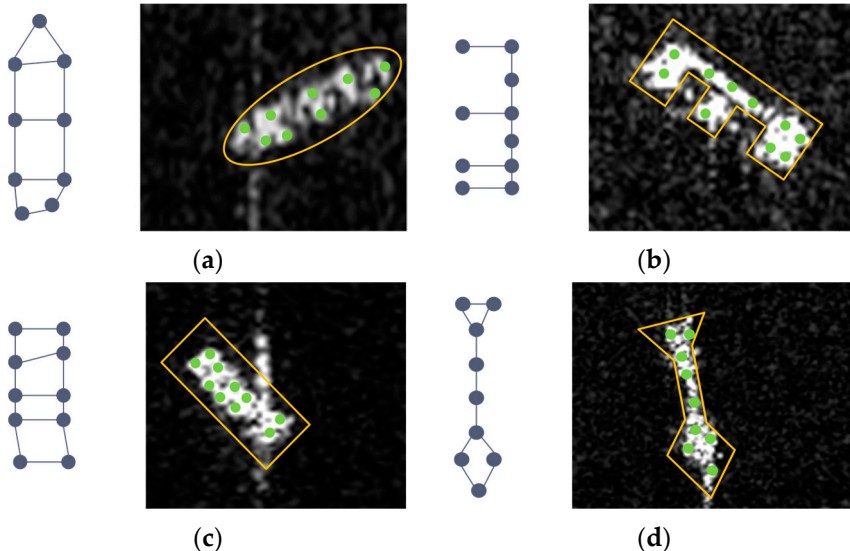

**Figure 2.** Irregular shape and topology of ship targets. The topology is represented by clusters of points, and the yellow lines represent the irregular shapes of ship targets. (**a**) represents an ellipse-like shape, (**b**) represents an E-like shape, (**c**) represents a rectangle-like shape, and (**d**) represents an arrow-like shape.

To overcome the above limitations, we proposed a novel morphology and topology-based feature alignment network (MT-FANet) for SAR ship target detection. First, we adopted deformable convolutions to cope with the irregular shape of the ship target in SAR images. Deformable convolutions can adaptively adjust the shape and position of the convolution kernel, thus more accurately extracting target morphology features. Second, we developed a morphological and topological feature pyramid network (MT-FPN), which combines the extracted target morphological features and dynamically allocates different weights based on the mutual relationships between scatter points at different locations. This process can capture the inherent topological structure information of ship targets, thereby achieving fewer missed detections and false alarms. Finally, we designed a rotation alignment feature head (RAFH), including prediction fine-tuning and feature differentiation, to solve issues such as feature misalignment and to implement rotated bounding box prediction. Our proposed method had better practicality and effectiveness in detecting irregularly shaped ship targets with complex backgrounds.

Our main contributions can be summarized as follows:

1. We adopted deformable convolutions to improve the network's feature representation ability for irregularly shaped ship targets, focusing more on the features of the target itself rather than the background, and thus mitigating the impacts of complex background interference.
2. It is well-known that the topological structures of ship targets contain important feature information. Therefore, we developed a novel morphology and topology

feature pyramid network (MT-FPN) to exploit the inherent topological structure information of SAR ship targets, which can elucidate effective features for consequent ship target detection.

3.    To achieve a balance between the speed and accuracy of the proposed detection model, a rotation alignment feature head (RAFH) was designed to predict fine-tuning and feature differentiation. This addresses the feature misalignment issue and enables rotation bounding box prediction, thus improving the model's detection performance.

The remaining sections of this paper are organized as follows. Section 2 discusses related work. In Section 3, we provide a detailed description of the proposed methods. The experimental results and ablation studies are presented in Section 4, and we further discuss the experimental results in Section 5. Finally, in Section 6, we conclude the paper.

## 2. Related Work

In this section, we introduce the development of deep learning-based detection algorithms for SAR ship targets and improvements in the feature pyramid structure.

### 2.1. Deep Learning Detection Method for SAR Ship Targets

Compared to traditional detection algorithms, CNNs can adaptively extract features and have better generalization ability [30]. SAR ship target detection based on deep learning has received widespread attention [32–34]. As shown in Table 1, recently, some research has focused on enhancing the performance of feature extraction [35–39]. Rostami et al. [36] proposed a framework based on domain-transferred knowledge to train from related domains where large amounts of data are readily available to achieve SAR images better generalization capabilities. It is also a good idea to increase the sample perspective in order to enhance feature extraction [38,39]. Lou et al. [38] used a knowledge transfer network to generate fake SAR images, then used both the fake image and the real image as the input of the ship detection network, which improved the generalization performance.

**Table 1.** Classification of methods to enhance feature extraction performance in SAR ship and object detection.

| Method | Obtained Results | Related References |
|---|---|---|
| Pre-training and transfer learning | | [35–37], etc. |
| Data augmentation | Mitigating limitations of fewer samples | [38,39], etc. |
| Feature selection | Enhanced model architecture | [22,40–42], etc. |

Some studies exploit target features to optimize feature extraction. Guo et al. [40] proposed a method of rotating Libra R-CNN to balance multiple semantic levels, including sample level, feature level, and object level, to solve the problem of the dense distribution of objects. Fu et al. [41] introduced a context-aware feature selection module to suppress the interference of the background, and defined a set of scattered key points to describe the characteristics of local scattering regions, thus improving the detection performance for complex scenes. Kang et al. [42] designed a scatter feature relational network using the scatter point relation module to realize the analysis and the association of scatter points to ensure the integrity of object detection. In addition, the fusion module and contextual feature attention were used to capture semantic and spatial information. With the rapid development of deep learning in the SAR field, many detectors based on oriented bounding boxes have been proposed [30,31,43]. Shao et al. [22] designed a rotation-balanced feature alignment network to accurately identify SAR ship targets. This method reduces the negative impact of multi-scale feature differences by balancing the attention pyramid, and uses a deformable convolutional network [31] to deal with feature misalignment so as to achieve accurate recognition of SAR ship targets.

Despite the significant achievements in SAR ship detection during our preliminary work, numerous challenges remain to be overcome in the detection of rotated ship targets in SAR. In the above method [41,42], only the relevance of ship target scattering points is used,

and information on the ship's target shape is not paid enough attention. The irregular shape and underutilization of the inherent topology of ship targets limit the feature extraction performance of existing models, resulting in limited detection performance. Therefore, in order to improve the effectiveness of ship target feature extraction and to achieve accurate detection of SAR ship targets in complex scenes, we propose a novel network architecture based on morphological and topological information.

### 2.2. Feature Pyramid Structure

Multi-scale detection is significant for network scale invariance in image detection. The effective fusion of features at different scales can promote the interaction of cross-scale information and improve the detection performance of the network [22–26,44–46]. FPN [44] is a pioneering work that includes a top-down path and skip connections, providing rich semantic information and multi-scale context to improve detection performance and handle objects at different scales. Fu et al. [45] found that different layers do not contribute equally to balancing semantic features when designing a feature balance and refinement network (FBR-Net). Therefore, they used level-based attention and spatial channel attention to adaptively learn the weight of fusion, so as to balance multiple features at different levels and improve the detection performance of SAR ship targets. The attention-guided balanced feature pyramid network (A-BFPN) [46] represents a further optimization of the FPN structure. The method uses an enhanced refinement module to improve balance and FPN's ability to represent ship objects after feature fusion. It also uses channel attention to guide the recovery of features at different levels, thereby reducing the feature overlap problem caused by fusion and improving detection performance. These methods balance the weights of semantic information and spatial location information at different levels by different means, thereby improving the accuracy of detection.

We hope to design a concise and effective FPN structure without being overly complex. Therefore, we propose a morphology and topology feature pyramid network (MT-FPN). The network uses known ship morphology and topology information to reconstruct features and builds a feature fusion module on this basis. MT-FPN makes full use of the target shape and inherent topological structure information to make the features of each level more robust. At the same time, the high-level semantic information is simply and effectively transferred through the top-down path, thereby improving the detection performance of the network.

## 3. Proposed Method Description

In this paper, we propose a morphology and topology-based feature alignment network (MT-FANet), which consists of three parts: a backbone network; a feature fusion network, which is the morphological and topological feature pyramid network (MT-FPN); and a detection head, which is the rotation alignment feature head (RAFH), as shown in Figure 3. The backbone network uses ResNet50 [47] to extract features from the original images. MT-FPN improves upon the FPN module in RetinaNet [23] by adding a designed morphology and topology module (MTM) for the purpose of strengthening the ship targets' morphology features and extracting topological information. RAFH includes rotation offset prediction for feature alignment and decoupled feature prediction for final classification and prediction. This section introduces the overall structure of MT-FANet, explains each module's functionality and characteristics, analyzes the network structure's important calculation method, and presents the loss function used in the training process.

### 3.1. Overview of the Proposed MT-FANet

In this section, we introduce the proposed MT-FANet, the architecture of which is shown in Figure 3. The proposed method is based on the single-stage detector RetinaNet [23], which has fewer network parameters and a faster speed than two-stage algorithms. Based on RetinaNet, we propose that the irregular shape of ship targets and the topological structure information between scatter points be used for ship targets to guide

feature extraction and fusion in the network and to improve its detection performance. Specifically, we first used ResNet50 [47] as the backbone to extract the raw features of the images. ResNet50 has 50 layers and comprises 5 convolutional stages. The outputs of the last three stages of the backbone network ResNet50, denoted as {C3, C4, C5}, had downsampling ratios of {8, 16, 32} relative to the input image and channel output sizes of {512, 1024, 2048}, and were used as the input features of the subsequent morphology and topology module (MTM).

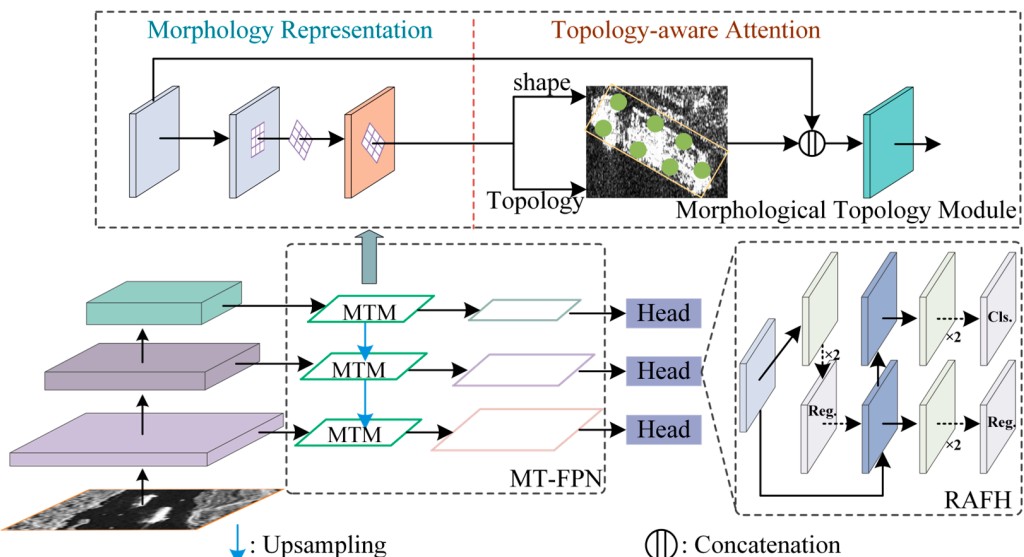

**Figure 3.** Framework of the proposed MT-FANet. It consists of three parts: the backbone network, the MT-FPN feature pyramid network, and the RAFH rotation alignment feature detection head. The sizes of the input and output feature maps of the MTM remain the same.

We designed a brand-new feature pyramid network, MT-FPN, to refine the features. This network uses MTM to enhance the representation ability with respect to the ship targets' morphologies and combines the more accurate morphology features to fully extract inherent topological information for the ship targets. MT-FPN also performs a simple fusion of the features from each layer, enhancing the interaction of information between each layer to improve the multi-scale detection performance. Finally, the prediction head, RAFH, uses a parameter-sharing mechanism and a two-stage prediction strategy, reducing the complexity of the network while achieving feature alignment and differentiation processing to optimize the network's performance further. In rotation offset prediction, we used the prediction of rotation box regression and the input feature map to align the features. In decoupled feature prediction, we refined the features to better complete the final classification and regression tasks. The anchor is represented by a 5-dimensional vector (x, y, w, h, θ) for both regression tasks, as shown in Figure 4. Here, (x, y) are the coordinates of the center point; (w, h) denotes the long and short sides of the rectangle; and θ is the angle between the positive x-axis and the long side, with a range of $[-\pi/4, \pi3/4)$, where clockwise is positive and counterclockwise is negative. The classification task distinguished between targets and backgrounds in the predicted boxes.

### 3.2. Morphology and Topology Feature Pyramid Network

In this section, we introduce the proposed MT-FPN network, which is a crucial module in MT-FANet for refining the feature maps of the ResNet50 backbone network. It can improve the detection performance of multi-scale irregular targets in complex backgrounds. MT-FPN comprises the morphology and topology module (MTM) and the feature pyramid network (FPN).

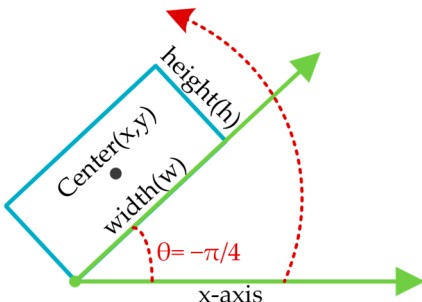

**Figure 4.** The physical meanings of each parameter of the rotated bounding boxes.

### 3.2.1. Feature Fusion

Three operations were performed in the morphology and topology feature pyramid network (MT-FPN). Firstly, the deformable convolution [31] in the morphology topology module (MTM) was utilized to extract the morphology feature of the target, thereby enhancing the network's capability to represent ship morphology. Then, by using the topology-aware attention in the morphology and topology module (MTM) and combining it with complete morphology features, the topological relationships between each scattering point in the ship target were calculated, thus completing the extraction of topological information among the scattering points in the ship target. Finally, the morphology and topology features at different levels were downsampled and added together for simple and effective cross-scale fusion, as shown in Figure 5. Previous studies [48] have suggested that in the remote sensing detection field, due to the size characteristics of the target, the P6 and P7 layers are redundant structures. In SAR ship detection, we also made corresponding operations. In the subsequent sections, we describe a burn experiment to prove that this conclusion applies to SAR ship detection.

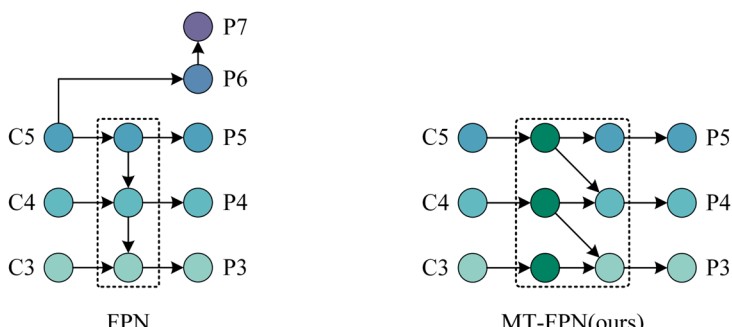

**Figure 5.** Structure of the feature pyramid network (FPN) and the morphology and topology feature pyramid network (MT-FPN).

### 3.2.2. Morphology and Topology Module

The detailed structure of the morphology and topology module (MTM) is shown in Figure 6, where "MR" represents morphology representation. In this section, we provide a detailed description of morphology representation, including the source of the morphology representation problem, available methods and measures, and implementation details of our approach.

Due to the different imaging mechanisms and conditions in SAR images [10], ship targets exhibit significant geometric variations (rotation, scaling, deformation, etc.), resulting in highly irregular target shapes. This also leads to the model paying excessive attention to background features. To overcome the influence of geometric transformations on the target images of the model, data augmentation [49] and DCN [31] can be used. Data augmentation generates more training samples by performing random operations, such as rotation, scaling, and translation, on the images. It provides the model with diverse samples during training, improving its generalization ability.

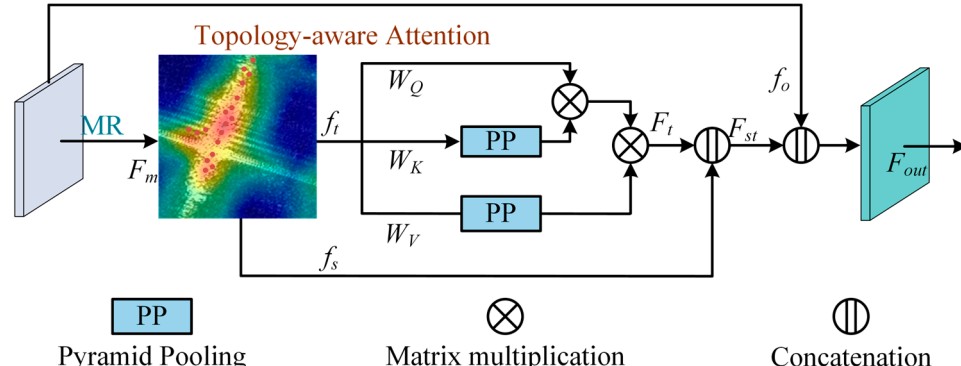

**Figure 6.** The architecture of the morphology and topology module (MTM). $W_Q$, $W_K$, and $W_V$, respectively, represent the weights of the $1 \times 1$ convolution operation, and their purpose is to project features into low dimensions; PP represents the global average pooling operation of multiple scales; concatenation represents channel splicing, and convolution is used for channel adjustment.

The enhancement of SAR images may yield unnatural data and cause model overfitting [49]. In addition, the shapes of ship targets in SAR images vary under different imaging conditions. Therefore, we used deformable convolution [31] to improve the ability to represent ship morphology after uniform channel adjustment (compressed to 128) on the output of the backbone network. This convolution was able to adaptively adjust the convolution kernel based on input features, thereby improving the detection performance of the model in the presence of rotation and multiscale targets and better capturing the diverse shapes of ship targets in the feature map. Specifically, deformable convolution calculates new sampling positions and weight coefficients through offset convolution for each feature point in the input feature map. Then, it performs convolution using the offset and weight. As shown in Figure 7, this convolution is implemented by adding a learnable offset layer before the traditional convolution layer, with a channel number of $3 \times$ *N, where N represents the number of convolution kernels of the deformable convolution. The output offset is used to adjust the sampling position of the convolution kernel in the input feature map, while the weight ensures the effectiveness of feature extraction. This method improves the robustness of the network to geometric transformations, better matches the irregular shapes of ship targets in the feature map, and enhances the network's representation ability for target morphology. The specific mathematical expression is shown below:

$$y(p) = \sum_{k=1}^{K} \omega_k \cdot x(p + p_k + \Delta p_k) \cdot \Delta m_k \tag{1}$$

where $x(\cdot)$ represents the location of the input feature map, the feature map is set to $H \times W \times 128$ $y(p)$ and represents the location of the output feature map, $\omega_k$ represents the weight of the convolution kernel, $\Delta p_k$ represents the offset of the sampling position, $K$ is the number of convolution kernels (usually taken as $3 \times 3$), $\Delta m_k$ is the weight coefficient of the sampling point, and the final output size remains as $H \times W \times 128$. Adding offset and weight to the original convolution operation increased our ability to explore target morphology.

In complex backgrounds, ship targets have highly similar features to their surrounding environments [22,29,30,41]. More than simply strengthening the target's morphology representation is required to achieve accurate detection. After the analysis of SAR images, ship targets are usually regarded as groups of continuous scattering points, and the arrangement of these scattering points and their relationships can provide inherent topological structure information about the ship target. For example, the rear of a ship is usually wider than the front, and some scattering points may concentrate in specific areas of the ship, such as the bow and stern. Therefore, we developed topological structure-aware attention to extract topological structure information from between the ship target's scattering points, thus identifying rich and novel topological features for the network. In previous stud-

ies, Hu et al. [50] have proposed an attention module derived from the Transformer [51] to describe pairwise relationships between targets. To better describe the topological structure features between the scattering points of ship targets, we designed topological structure-aware attention in MTM, as shown in Figure 6. We applied the self-attention mechanism to extract topological structure information, assuming that the feature $F_m$ after shape representation consists of shape features topological features to be extracted. The feature $F_m$ underwent feature projection to obtain the easy-to-use shape feature $f_s$ and the to-be-extracted topology feature $f_t$. Then, the relative distances between scattering points were calculated for ship targets based on the feature $f_t$ and these distances were used as attention weights to weigh the features and obtain a feature representation that contained topological structure information.

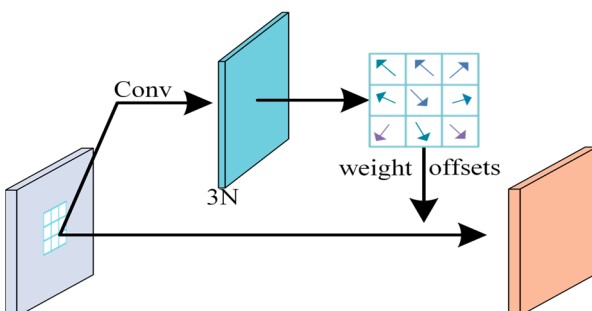

**Figure 7.** Architecture of deformable convolution. Conv represents a $3 \times 3$ convolution operation, and the number of output channels is $3 \times N$.

The self-attention mechanism can effectively capture topological structure features between scattering points of ship targets. In scaled dot-product attention, given an input consisting of queries, keys, and values represented by the matrices **Q**, **K**, and **V**, respectively, the output value can be efficiently computed using the following formula:

$$\boldsymbol{\upsilon}_{out} = \text{softmax}\left(\frac{\mathbf{Q}^T\mathbf{K}}{\sqrt{d_k}}\right)\mathbf{V} \tag{2}$$

where **Q** and **K** are computed along the dimension of $d_k$ and normalized by a scale factor $\sqrt{d_k}$ to regulate the magnitude of attention weights. In our work, to reduce the computational burden of matrix multiplication, we adopt a method inspired by [52] to simplify the computation of self-attention using the following expression:

$$\mathbf{K}_{(d_k \times N)} = [\text{GAP}_1(\mathbf{K}), \text{GAP}_3(\mathbf{K}), \text{GAP}_6(\mathbf{K}), \text{GAP}_8(\mathbf{K})] \tag{3}$$

$$\mathbf{V}_{(d_k \times N)} = [\text{GAP}_1(\mathbf{V}), \text{GAP}_3(\mathbf{V}), \text{GAP}_6(\mathbf{V}), \text{GAP}_8(\mathbf{V})] \tag{4}$$

where **K** and **V** are obtained by $1 \times 1$ convolution encoding and dimension reduction, resulting in matrices with sizes $(d_k, H \times W)$. Here, $d_k$ is the number of channels (we set it to 64) and $H \times W$ denotes the height and width of the feature map. $\text{GAP}_n(\cdot)$ denotes global average pooling, with an output size of $n \times n$. Using the idea of pyramid pooling, representative points are sampled from different scales and then concatenated to obtain $\mathbf{K}_{(d_k \times N)}$ and $\mathbf{V}_{(d_k \times N)}$, where $N = 1 \times 1 + 3 \times 3 + 6 \times 6 + 8 \times 8 = 110$ and $[\cdot]$ represents concatenation.

The above equation effectively captures the topological structure information between scattering points and enables the extraction of topological features. Specifically, it computes the similarity (position relation) between each scattering point and all other scattering points to determine its weight, which is then applied to calculate the output of that point. The extracted topological features, denoted as $F_t$, are concatenated with the shape feature $f_s$ as follows:

$$F_{st} = conv([F_t, f_s]) \tag{5}$$

where the $conv(\cdot)$ operation refers to a $1 \times 1$ convolution (input channel = 256, output channel = 128), which aims to keep the number of channels in the output feature $F_{st}$ consistent with that in the input feature $f_s$. The obtained feature $F_{st}$ contains rich information on the shape and topology of the target, which can effectively enhance the feature representation.

After obtaining the feature $F$ from the MTM structure, in order to ensure that the features extracted by the backbone network were fully utilized, we performed a cascading operation to aggregate $F$ with the original feature map $f_o$, then used the result for feature pyramid network fusion:

$$F_{out} = conv([f_o, F]) \tag{6}$$

where $conv(\cdot)$ operation refers to a $1 \times 1$ convolution, it has 256 input channels, and the number of output channels is 256. This aims to keep the number of channels in the output feature $F_{out}$ consistent with that in the input feature $f_o$ for the purpose of facilitating subsequent feature fusion.

In MTM, deformable convolution greatly enhances adaptability to the morphological variations of the ship targets. Meanwhile, the use of topology-aware attention leads to the deconstruction of morphology features and the extraction of topological structure information. This enables the network to better focus on the morphological and topological structural features of the target itself with complex backgrounds, thus achieving a deeper understanding of the morphology and inherent topology structure of the ship target, and better suppressing false negatives and false positives in the network.

### 3.3. Rotation Alignment Feature Head

In horizontal bounding-box detection networks, convolutional features and horizontal anchors are aligned, making it easy to predict bounding boxes [32]. However, in rotated anchor networks, the asymmetric nature of the rotated boxes causes misalignment between convolutional features and anchors. To solve this problem, we developed a rotation alignment feature head (RAFH), which adopted a two-stage prediction strategy. We first made regression predictions, then used the first regression predictions to align and sample features, and finally performed the classification and regression predictions. The detection head adopted a parameter-sharing mechanism [24] where three output scales shared one detection head, significantly reducing the model parameters. This section explains the first regression prediction part (rotation offset prediction) and the second precise prediction part (decoupled feature prediction). Compared with the baseline's decoupled head, our RAFH detection head only adds a few extra parameters to achieve feature alignment and second-stage prediction while omitting the classification subnet in the first stage, making our network lighter and more accurate.

#### 3.3.1. Rotation Offset Prediction

We design a rotation offset prediction structure to address the misalignment between rotated bounding boxes and axis-aligned convolution features in SAR ship detection [33,34]. This structure used regression prediction to refine features for more accurate prediction in the subsequent decoupled feature prediction. Specifically, we first performed regression prediction on the aforementioned MT-FPN's features ($H \times W \times 256$); then used the prediction result to project the offset adjustment information sampled by convolution; and, finally, aligned the input feature map $f_i$. As shown in Figure 8, this structure only had a regression prediction branch, and the convolution layer of the branch was consistent with the baseline setting. Since we used the five-parameter definition method for rotated boxes, the regression output was a feature map with a size of $H \times W \times 5$. Through decoding, we were able to obtain the coarse anchor boxes for each feature point, and then to map them to the required sampling offset information. The specific formula for calculating the offset $O$ was as follows:

$$O = \left\{ \mathbf{L}_{P_o}^{P_k} - \mathbf{P}_k \right\} \tag{7}$$

$$\mathbf{L}_{P_o}^{\mathbf{P_k}} = \frac{1}{S}(\frac{1}{K}(w,h) \cdot \mathbf{P_k} \cdot \mathbf{R}^{\mathrm{T}}(\theta)) \tag{8}$$

where $O$ represents the offset value; $\mathbf{L}_{P_o}^{\mathbf{P_k}}$ represents the new sampling position obtained by projecting the sampling kernel of the sampling position $P_0$ using the predicted anchor box; $S$ represents the stride of the feature map; $K$ represents the size of the convolution kernel, which was set to 3 in our case; $\mathbf{P}_K$ represents the grid coordinates of the original sampling position, which belonged to $\{(-1, -1), (-1, 0), \dots, (0, 1), (1, 1)\}$, according to the convolution size; $\mathbf{R}^{\mathrm{T}}(\theta) = (cos, -sin; sin, cos)^{\mathrm{T}}$ represents a rotation matrix used for horizontally projecting the new sampling position.

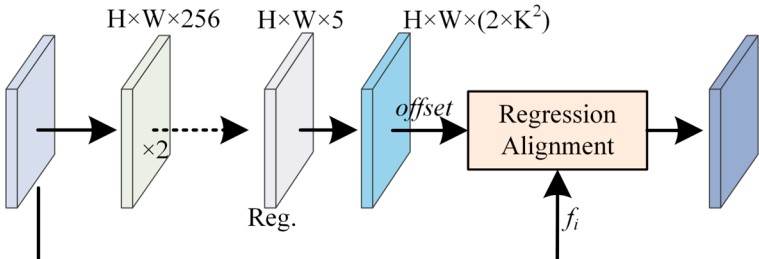

**Figure 8.** Architecture of the rotation offset prediction. Reg. represents the regression prediction module.

After obtaining the offset information, the feature alignment operation was performed by inputting the feature $f_i$ and offset into the regression alignment layer. It is worth noting that the output size of the regression alignment layer remained as $H \times W \times 256$. For each point, the feature alignment result $\mathbf{F}_A$ was the product of the learnable convolution weights and the feature at the new sampling position. The calculation formula was as follows:

$$\mathbf{F}_A = \sum_{o \in O} \omega(\mathbf{P}_k)X(\mathbf{P}_k + \mathbf{o}) \tag{9}$$

where $\omega(\cdot)$ represents the convolution kernel weight, and $X(\cdot)$ represents the sampled value in the feature map.

### 3.3.2. Decoupled Feature Prediction

This section introduces the decoupled feature prediction in the RAFH detection head, which is used for detecting and classifying ship targets in SAR images. Due to the difference in required features for regression and classification tasks, conflicts may arise [24]. To overcome this issue, we used a spatial decoupling method to separate features and to complete target boundary regression and classification using regression and classification subnets, as shown in Figure 9. Specifically, for the input that underwent feature alignment, we use active rotation convolution [53] for direction encoding to further enhance the direction perception of ship targets. This explicitly encodes the feature perception of the target direction on the channel. For the input feature map $\mathbf{F}$, the output $\mathbf{F}_o$ in the $i$ direction can be represented as:

$$\mathbf{F}_o{}^{(i)} = \left\{ \boldsymbol{\omega}_{\theta_i}^{(k)} \cdot \mathbf{F}^{(k)} \right\}_{0,1,\cdots,K-1}, \theta_i = i\frac{2\pi}{N}, i = 0, \dots, N-1 \tag{10}$$

where $\theta_i$ represents the angle information encoded in clockwise manner; $\boldsymbol{\omega}_{\theta_i}^{(k)} \cdot \mathbf{F}^{(k)}$ indicates the calculation between the feature map $\mathbf{F}$ and the rotation convolution kernel $\boldsymbol{\omega}$ in the $k$-th channel; and $K * N$ should be equal to the number of channels of the input feature $\mathbf{F}$. For example, if the channel number is 256 and $N$ is set to 8, then $K$ would be 32. Obtaining orientation-aware features will help to improve the accuracy and precision of bounding box regression. Secondly, we preferred to obtain features invariant to the direction for the classification task. To achieve this, we chose the global average pooling of each encoding

channel to extract invariant features, realizing the differentiation of classification and regression features. The formula was as follows:

$$\tilde{\mathbf{F}} = \text{mean}(\mathbf{F}^{(n)}), 0 < n < N - 1 \tag{11}$$

where $\mathbf{F}^{(n)}$ represents each encoding channel and the size of $\mathbf{F}^{(\cdot)}$ is $H \times W \times 32$. Finally, the features with direction sensitivity and invariant features were fed into the regression and classification subnetworks, respectively, to obtain the target box prediction output ($H \times W \times 5$) and classification prediction output ($H \times W \times 1$).

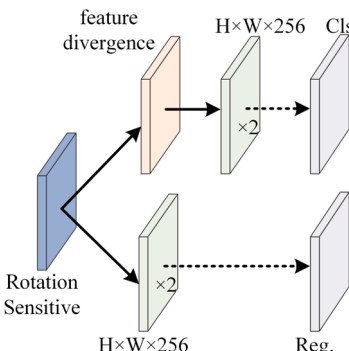

**Figure 9.** Architecture of decoupled feature prediction. Rotation sensitive means explicitly encoding the angle information on the feature channel; feature divergence means merging the angle information on the channel; cls. represents the classification subnetwork; reg. represents the regression subnetwork.

*3.4. Loss Function*

MT-FANet uses a type of multi-task loss that includes the rotation offset prediction and decoupled feature prediction losses. The rotation offset prediction loss only consists of a regression loss, while the decoupled feature prediction loss has both a classification loss and a regression loss. The multi-task loss function is defined as follows:

$$L = L_R + L_D \tag{12}$$

$$L_R = \lambda \frac{1}{N_R} \sum_i p_i^* L_{reg}(t_i, t_i^*) \tag{13}$$

$$L_D = \frac{1}{N_D} \left( \sum_i L_{cls}(p_i, p_i^*) + \lambda \sum_i p_i^* L_{reg}(t_i, t_i^*) \right) \tag{14}$$

where $N_R$ and $N_D$ are the number of positive samples in rotation offset prediction and decoupled feature prediction, respectively; $p_i^*$ equals 1 if sample $i$ is positive, while otherwise it equals 0; $p_i$ represents the ground truth label of anchor $i$; focal loss [23] is used for classification loss $L_{cls}$; $t_i$ represents the offset between anchor and ground truth; $t_i^*$ represents the offset between prediction and ground truth; smooth L1 [17] is used for regression loss $L_{reg}$; and $\lambda$ is a hyperparameter for balancing classification and regression losses, usually set to 1. However, in our work, the loss function contained a classification loss and two regression losses. Thus, we set $\lambda = 0.5$ for simple balance. In follow-up experiments, we intend to provide ablation experiments to verify the effectiveness of the hyperparameter $\lambda$ setting.

## 4. Experimental Results

In this section, we describe the experiments performed to validate the performance of the proposed MT-AFNet for ship detection in SAR images. Further, we demonstrate the effectiveness of the designed MTM and RAFH. First, we introduce the RSDD-SAR

dataset [16] used in this paper, the evaluation metrics, and the experimental details. Then, we report the advanced performance of our MT-AFNet compared to existing methods on the RSDD-SAR dataset. Finally, we describe the ablation studies which were used to evaluate the impact of each module on performance.

### 4.1. Experimental Datasets and Details
4.1.1. Datasets

The dataset used in this paper was the RSDD-SAR dataset [16], which currently has the largest number of samples and the richest scenes with respect to SAR ship rotation detection. It contains 7000 images and 10,263 ship instances. The dataset includes images of multiple imaging modes, polarization modes, and resolutions. The specific data statistics are shown in Table 2.

**Table 2.** Basic information of RSDD-SAR dataset.

| Parameter | Value |
|---|---|
| Number of images | 7000 |
| Image size | 512 × 512 |
| Number of trains | 5000 |
| Number of tests | 2000 |
| Polarization | HH, HV, VH, DH, DV, VV |
| Imaging mode | SM, FSII, FSI, QPSI, UFS, SS |
| Resolution | 2~20 m |

We display the ship targets' rotation angles and aspect ratios in Figure 10a,b. The rotation angles ranged from $-\pi/4$ to $\pi/2$, with fewer instances between $\pi/2$ and $3\pi/4$, and the distribution was generally uniform. The aspect ratios were mainly between 1.5 to 7.5. These values suggest that the targets have diverse rotations and significant aspect ratios.

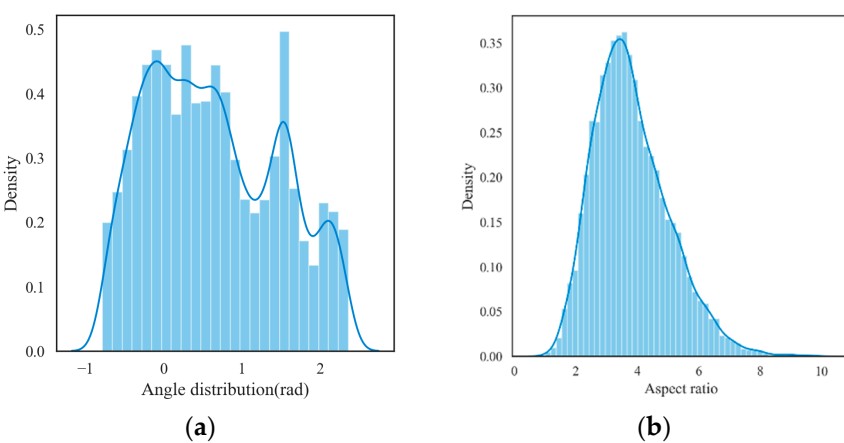

(**a**)  (**b**)

**Figure 10.** Distribution of rotation angles and aspect ratios of ship targets in the RSDD-SAR dataset: (**a**) represents the distribution map of ship angles; (**b**) represents the aspect ratio distribution map of the ships.

The images in Figure 11 illustrate the RSDD-SAR dataset, which encompasses a diverse range of ship target scenes with a wide range of scales. In inshore environments, ship targets are frequently blended into intricate backgrounds, and it is possible for them to be arranged densely, making their precise detection and localization more challenging. Therefore, the introduction and utilization of this dataset is highly significant for advancing and investigating ship target detection algorithms.

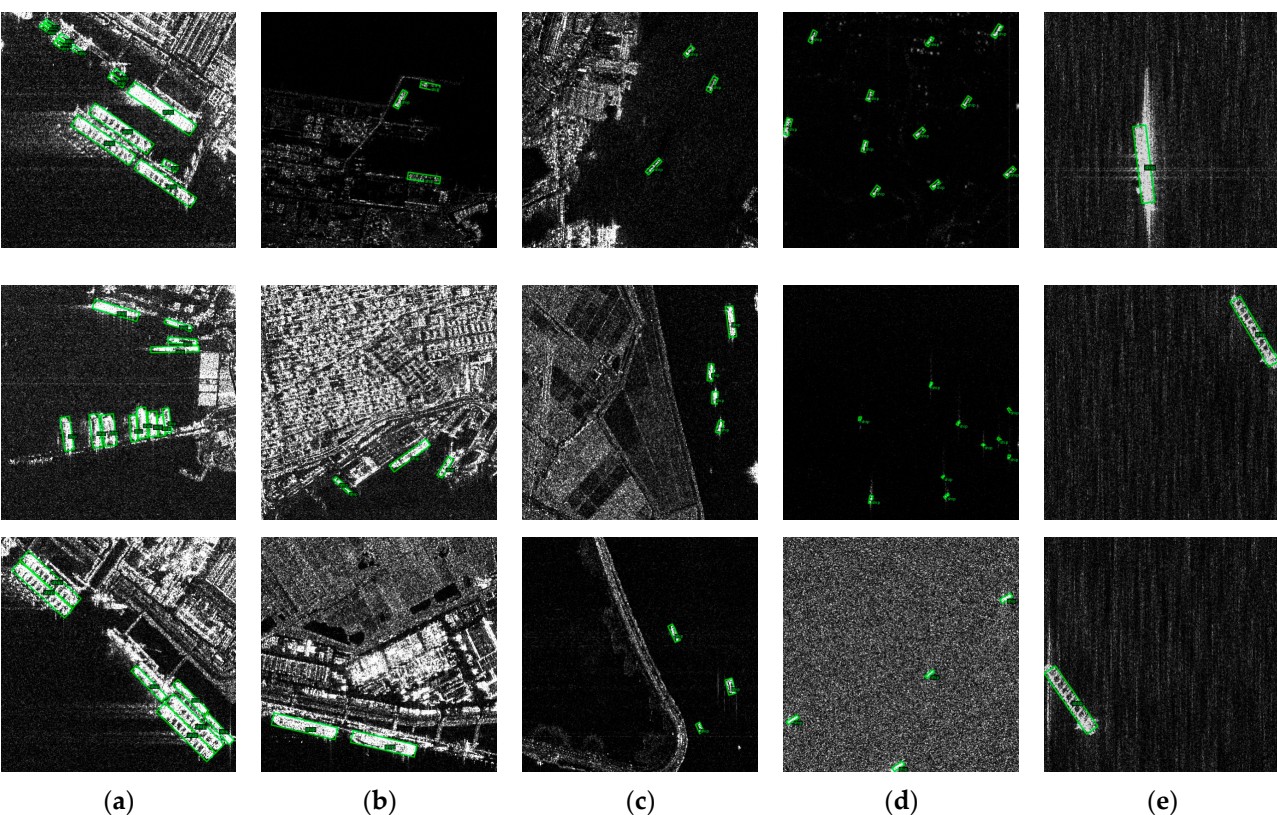

**Figure 11.** RSDD-SAR dataset examples of various scenes: (**a**) shows the dense arrangement scene; (**b**) shows the port scene; (**c**) shows the channel scene; (**d**) shows the low-resolution scene; and (**e**) shows the high-resolution scene.

### 4.1.2. Experimental Details

We used Resnet50 [47], pre-trained on ImageNet, as the backbone network for MT-FANet and other comparative algorithms, unless there were special experimental instructions. Data augmentation included horizontal random flipping with a probability of 0.5. We used SGD as the optimizer, with an initial learning rate of 0.0025, a momentum of 0.9, and a weight decay of 0.0001. We used L2 norm gradient clipping to increase the model's training stability, with a maximum gradient norm of 35. The iteration was set to 12, and the learning rate was reduced by a factor of 10 at the 7th and 10th iterations. The first 500 batches used a linear warm-up learning rate, with an initial warm-up learning rate of one-third of the initial learning rate. The batch size during the training stage was set to 4. The algorithm's training and testing image sizes were unified at $512 \times 512$, and the confidence threshold for testing was set to 0.05. All experiments were repeated five times, and the results are presented as mean and standard deviation. When discussing the experiment, we mainly describe the mean change. We implemented the algorithm using the PyTorch and MMDet toolbox and tested it on a personal computer with an Intel Core i5-11400F CPU and an NVIDIA GeForce RTX3060 GPU. We published the code at https://github.com/CQUSARDet/MT-FANet (accessed on 29 May 2023).

### 4.2. Evaluation Metrics

We used multiple evaluation metrics to assess the performance of the network, including network parameters (Params) such as the number of floating-point operations (FLOPs), inshore $AP_{50}$, offshore $AP_{50}$, recall, F1 score, and overall $AP_{50}$. $AP_{50}$ represents the average precision (AP) when the intersection over union (IoU) threshold is 0.5. The higher the AP

value, the higher the network detection accuracy. To calculate AP, we first need to calculate recall and precision. Precision and recall can be defined as:

$$\text{Precision} = \frac{\text{TP}}{\text{TP} + \text{FP}} \tag{15}$$

$$\text{Recall} = \frac{\text{TP}}{\text{TP} + \text{FN}} \tag{16}$$

where TP is the number of ships detected correctly, FP is the number of false positives, and FN is the number of ships missed. The formula for calculating AP was as follows:

$$\text{AP} = \int_0^1 P(R)dR \tag{17}$$

where $P$ represents precision and $R$ represents recall. $P(R)$ represents the precision–recall curve.

The F1 score takes into account the precision and recall of the model, and can be calculated by the following formula:

$$\text{F1-score} = 2 \times \frac{\text{Precision} \times \text{Recall}}{\text{Precision} + \text{Recall}} \tag{18}$$

*4.3. Comparison with State-of-the-Art Methods*

To validate the superiority of our algorithm, we compared it with seven other rotation detection algorithms on the RSDD-SAR dataset, as shown in Table 3 The detection results of the different methods are taken from reference [16]. MT-FANet outperformed the seven state-of-the-art methods based on the two-stage, one-stage, and anchor-free algorithms. Regarding overall AP50, MT-FANet performed 1.48% better than the second-best algorithm, while in inshore and offshore AP50, it performed 0.36% and 1.19% better, respectively. It is worth noting that the second-best algorithm was different for each metric. Additionally, MT-FANet had the second-lowest number of model parameters and the lowest number of floating-point operations. These results indicate that MT-FANet fully considers the morphology representation of ship targets and the extraction of topological information, achieving a state-of-the-art performance in SAR ship target detection. Furthermore, it exhibits outstanding performance in recall, F1 score, model parameters, and floating-point operations, proving its superiority in practical applications.

**Table 3.** Comparison of detection results for different algorithms in the RSDD-SAR dataset.

| Method | $\text{AP}_{50}$ (%) | Recall (%) | F1 | O. $\text{AP}_{50}$ (%) | I. $\text{AP}_{50}$ (%) | Params (M) | FLOPs (G) |
|---|---|---|---|---|---|---|---|
| R-FasterR-CNN [17] | $83.44 \pm 0.34$ | $86.93 \pm 0.19$ | $85.15 \pm 0.26$ | $90.47 \pm 0.40$ | $49.44 \pm 0.51$ | 41.41 | 50.38 |
| RoI Transformer [32] | $88.39 \pm 0.02$ | $89.95 \pm 0.02$ | $89.17 \pm 0.01$ | $\underline{94.53 \pm 0.17}$ | $60.19 \pm 0.56$ | 55.32 | 51.48 |
| Oriented R-CNN [19] | $88.69 \pm 0.29$ | $90.50 \pm 0.23$ | $89.59 \pm 0.26$ | $90.56 \pm 0.30$ | $65.73 \pm 0.28$ | 41.35 | 50.41 |
| R-FCOS [26] | $85.35 \pm 0.13$ | $87.60 \pm 0.13$ | $86.46 \pm 0.12$ | $92.94 \pm 0.13$ | $50.12 \pm 0.45$ | **32.17** | 51.73 |
| CFA [33] | $\underline{89.36 \pm 0.09}$ | $\underline{91.50 \pm 0.39}$ | $\underline{90.41 \pm 0.23}$ | $90.80 \pm 0.32$ | $\underline{66.51 \pm 0.17}$ | 36.83 | $\underline{48.58}$ |
| R3Det [43] | $80.58 \pm 0.34$ | $82.88 \pm 0.14$ | $81.77 \pm 0.25$ | $89.76 \pm 0.46$ | $56.47 \pm 0.39$ | 41.81 | 83.91 |
| S2ANet [34] | $87.84 \pm 0.14$ | $89.17 \pm 0.19$ | $88.50 \pm 0.16$ | $93.31 \pm 0.16$ | $63.32 \pm 0.17$ | 36.45 | 49.40 |
| **Proposed method** | $\mathbf{90.84 \pm 0.18}$ | $\mathbf{92.21 \pm 0.21}$ | $\mathbf{91.52 \pm 0.22}$ | $\mathbf{95.72 \pm 0.19}$ | $\mathbf{66.87 \pm 0.39}$ | $\underline{33.73}$ | **43.96** |

The best-performing detection algorithm for each metric is shown in bold, while the second-best-performing algorithm is underlined. In all tables, "O." stands for offshore and "I." stands for inshore.

Figure 12 shows the detection results of our method compared to other methods, which performed second-best or best in different scenes and on different scales. From the figure, it can be seen that compared to methods such as R-FCOS [26], RoI Transformer [32], and CFA [33], our method, MT-FANet, focuses on the morphology and topology of the ship itself rather than its background features, resulting in better performance in terms of

suppressing false alarms and missed detections in SAR images. In addition, our method demonstrates better localization accuracy for densely arranged ship targets.

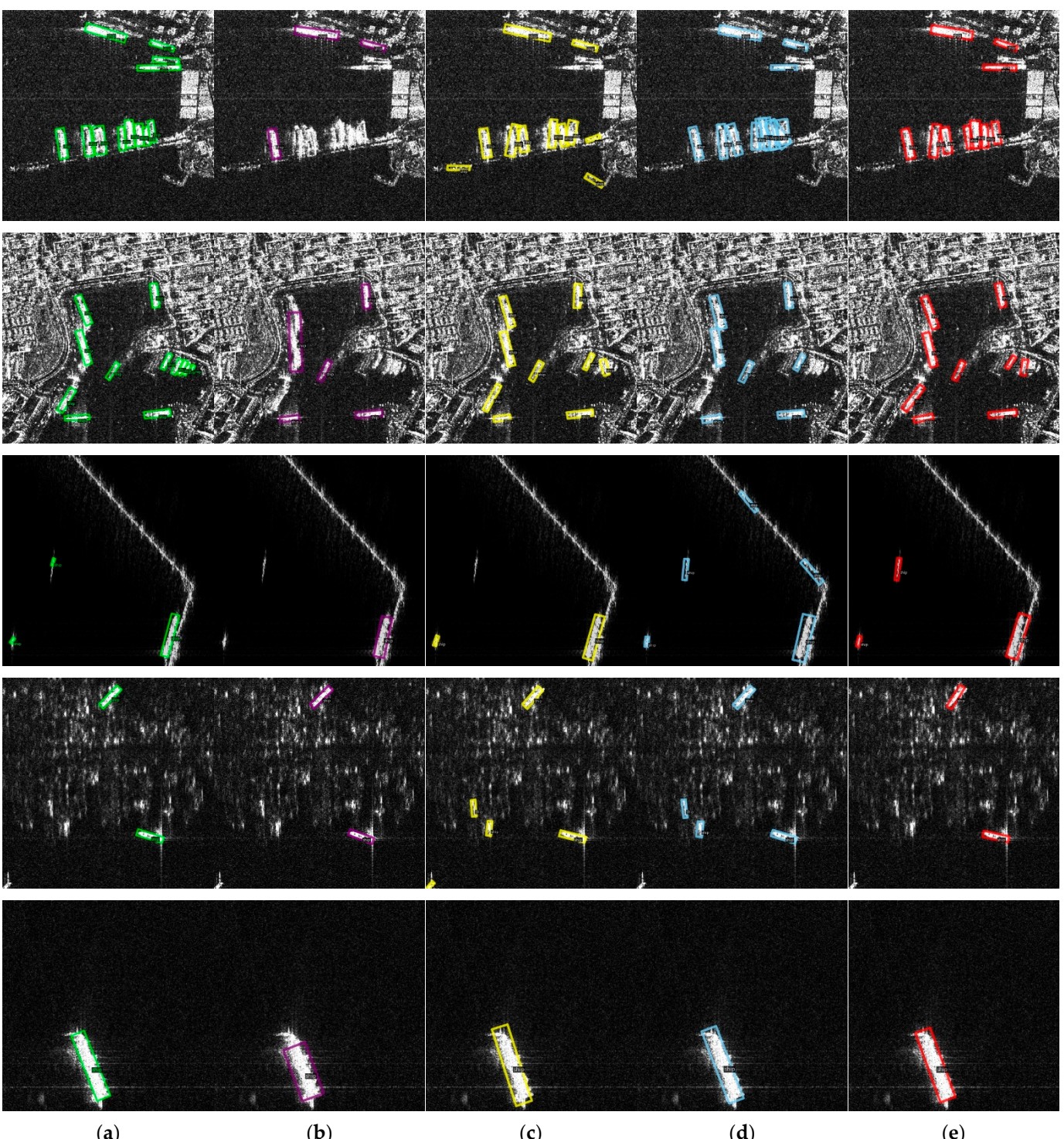

**Figure 12.** Detection results using different algorithms. (**a**) Ground truth; (**b**) results of R-FCOS; (**c**) results of RoI Transformer; (**d**) detection results of CFA; (**e**) results of our method, MT-FANet.

### 4.4. Ablation Studies

In this section, we describe the ablation experiments which were conducted on two critical areas requiring improvement in MT-FANet: the morphology and topology feature pyramid network (MT-FPN) and the rotation alignment feature head (RAFH). We quantita-

tively analyze the contributions of each module to MT-FANet and qualitatively analyze the advantages of our method compared to the baseline.

In Section 3.2.1, we mentioned that some structures in FPN were redundant for SAR ship detection. To validate this, we conducted a quantitative experiment on the RSDD-SAR test dataset, as shown in Table 4. Specifically, we removed the output layers P6 and P7 from RetinaNet and obtained a detector called "Modified-Baseline". We found that the "Modified-Baseline" achieved an AP50 of 84.28% in detection performance and reduced the number of floating-point operations and parameters (from 52.39 G to 51.7 G and from 36.13 M to 30.82 M, respectively). It also improved slightly in terms of recall and F1-score. Therefore, we used the "Modified-Baseline" as a new reference line for subsequent ablation studies.

**Table 4.** Analysis of different baseline architectures on RSDD-SAR.

| Method | F.P.L | $AP_{50}$ (%) | R (%) | F1 | O. $AP_{50}$(%) | I. $AP_{50}$ (%) | P. (M) | Fs. (G) |
|---|---|---|---|---|---|---|---|---|
| Baseline | P3~P7 | $83.97 \pm 0.10$ | $88.34 \pm 0.10$ | $86.10 \pm 0.15$ | $90.64 \pm 0.15$ | $54.62 \pm 0.18$ | 36.13 | 52.39 |
| Modified-Baseline | P3~P5 | $84.28 \pm 0.13$ | $88.82 \pm 0.17$ | $86.49 \pm 0.18$ | $90.83 \pm 0.14$ | $55.55 \pm 0.20$ | 30.82 | 51.70 |
| **Proposed method** | P3~P5 | $\mathbf{90.84 \pm 0.18}$ | $\mathbf{92.21 \pm 0.21}$ | $\mathbf{91.52 \pm 0.22}$ | $\mathbf{95.72 \pm 0.19}$ | $\mathbf{66.87 \pm 0.39}$ | **33.73** | **43.96** |

"F.P.L." refers to the number of levels in the feature pyramid. "R" refers to the recall. "P." refers to the params. "Fs." refers to the FLOPs. Bold text indicates the proposed method and its metrics in this and subsequent tables

Table 5 shows the quantitative impact of each critical improvement on detection accuracy. This experiment aimed to evaluate the performance improvement of the MT-FANet model, including the design of the MT-FPN module and the rotation alignment feature head (RAFH). In the table, "√" indicates the presence of an improved module, and "×" indicates the absence of the module. The results show that with the addition of improvements, the detection accuracy of the network gradually improved. In the SAR ship target detection task, compared with the baseline network, the $AP_{50}$ of MT-FANet increased by 6.56% (from 84.28% to 90.84%) and the offshore $AP_{50}$ increased by 4.89% (from 90.83% to 95.72%). The detection effect of inshore $AP_{50}$ improved even more significantly, with an increase of 11.32% (from 55.55% to 66.87%). In addition, we further illustrated the effectiveness of each step in the proposed method by providing the visualization results corresponding to each step. Figure 13 demonstrates the gradual improvement in missed and false detections of ship targets by introducing each essential improvement. Our method performed best in the detection of results, as shown in Figure 13d.

**Table 5.** Ablation study on various improvements proposed in MT-FANet on RSDD-SAR dataset.

| MT-FPN | RAFH | $AP_{50}$ (%) | Recall (%) | F1 | O. $AP_{50}$ (%) | I. $AP_{50}$ (%) | Params (M) | FLOPs (G) |
|---|---|---|---|---|---|---|---|---|
| × | × | $84.28 \pm 0.13$ | $88.82 \pm 0.17$ | $86.49 \pm 0.18$ | $90.83 \pm 0.14$ | $55.55 \pm 0.20$ | 30.82 | 51.70 |
| √ | × | $87.32 \pm 0.20$ | $89.13 \pm 0.18$ | $88.22 \pm 0.14$ | $92.90 \pm 0.21$ | $59.20 \pm 0.33$ | 33.68 | 53.62 |
| √ | √ | $\mathbf{90.84 \pm 0.18}$ | $\mathbf{92.21 \pm 0.21}$ | $\mathbf{91.52 \pm 0.22}$ | $\mathbf{95.72 \pm 0.19}$ | $\mathbf{66.87 \pm 0.39}$ | **33.73** | **43.96** |

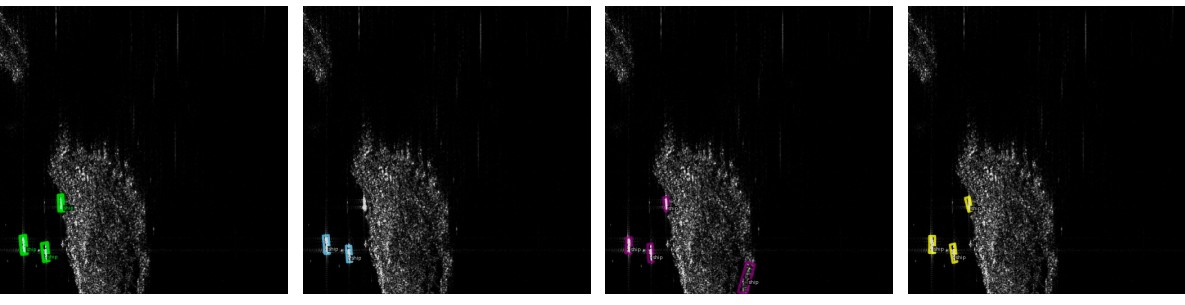

**Figure 13.** *Cont*.

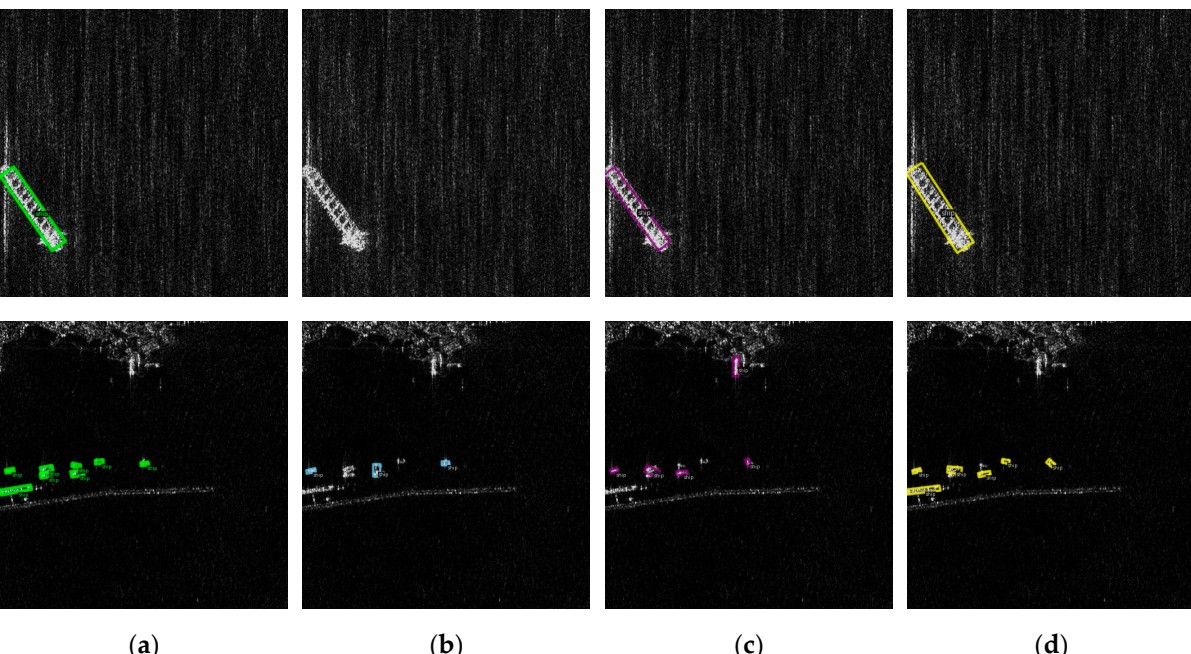

|  (**a**)  |  (**b**)  |  (**c**)  |  (**d**)  |

**Figure 13.** Visualization results of various proposed improvements. (**a**) Ground truth; (**b**) results without MT-FPN and RAFH; (**c**) results with MT-FPN; (**d**) results of proposed method.

The MT-FPN module enhanced the morphology representation of ship targets and enabled the extraction of topological information, allowing the network to more fully extract the features of ship targets and, thereby, to more successfully suppress false positives and missed detections of SAR images. The rotation alignment feature head efficiently achieved feature alignment and target prediction through two-stage prediction and feature separation strategies. It greatly reduced the model's floating-point operation count (from 51.70 G to 43.96 G) through its parameter-sharing mechanism, which improved its detection performance and efficiency.

When conducting ablation experiments on the MT-FPN module, we compared two identical networks, both using RAFH. The only difference was their feature refinement methods. One used traditional FPN, while the other used MT-FPN, which we proposed. Table 6 shows the ablation results of MT-FPN, demonstrating that the MT-FPN module can significantly improve detection accuracy. This shows that in MT-FPN, the morphological representation of the ship target is enhanced and the inherent topological information of the ship target is effectively utilized. As a result, the detection accuracy of the ship target was significantly improved.

**Table 6.** A study on the effectiveness of MT-FPN on RSDD-SAR.

| Method | $AP_{50}$ (%) | Recall (%) | F1 | O. $AP_{50}$ (%) | I. $AP_{50}$ (%) | Params (M) | FLOPs (G) |
|---|---|---|---|---|---|---|---|
| FPN | $88.64 \pm 0.21$ | $90.34 \pm 0.25$ | $89.47 \pm 0.22$ | $92.66 \pm 0.28$ | $58.42 \pm 0.24$ | 30.86 | 42.03 |
| **Proposed method** | **$90.84 \pm 0.18$** | **$92.21 \pm 0.21$** | **$91.52 \pm 0.22$** | **$95.72 \pm 0.19$** | **$66.87 \pm 0.39$** | **33.73** | **43.96** |

To efficiently perform object classification and location regression simultaneously, we balanced classification loss and regression loss in the loss function design. We set the hyperparameter $\lambda$ to 0.5 to avoid overweighting the regression loss. Through experimental verification, we found that when the network used MT-FANet, after setting $\lambda$ from 1 to 0.5, all evaluation indicators were improved (see Table 7). This shows that setting $\lambda$ to 0.5 in the proposed method can achieve a better balance between object classification and location regression, thus improving the overall detection performance.

**Table 7.** A study on the effectiveness of hyperparameter λ settings on RSDD-SAR.

| Hyperparameter λ set | AP$_{50}$ (%) | Recall (%) | F1 | O. AP$_{50}$ (%) | I. AP$_{50}$ (%) |
|---|---|---|---|---|---|
| λ = 1 | 90.19 ± 0.26 | 91.71 ± 0.24 | 90.94 ± 0.24 | 95.52 ± 0.32 | 63.54 ± 0.18 |
| **Proposed method (λ = 0.5)** | **90.84** ± 0.18 | **92.21** ± 0.21 | **91.52** ± 0.22 | **95.72** ± 0.19 | **66.87** ± 0.39 |

In Figure 14, we show the heat maps of the baseline and proposed MT-FANet in order to analyze and understand the advantages of our method qualitatively. We extracted the P3-level features of the MT-FPN and baseline FPN, and then performed max pooling on them to compress the channels. Next, we visualized the values and proportionally mixed them with the original images. The figure shows that MT-FANet can more successfully focus the features on the target morphology while effectively suppressing attention to interfering noise. This is mainly because our method focuses on representing the morphology of the ship target, and the extracted features can be adaptively mapped to obtain a more accurate shape of the ship target. In the case of the inshore, due to the high level of similarity between the complex background and the ship target, the baseline method cannot effectively extract features. This is shown in Figure 14b, where the generated heat map shows no significant difference in response to the ship or the noise. In contrast, as shown in Figure 14c, MT-FANet uses the inherent topological information of ship targets to guide the network model for feature selection, and can successfully suppress the network's attention to similar noise, thus enhancing the response to ship targets.

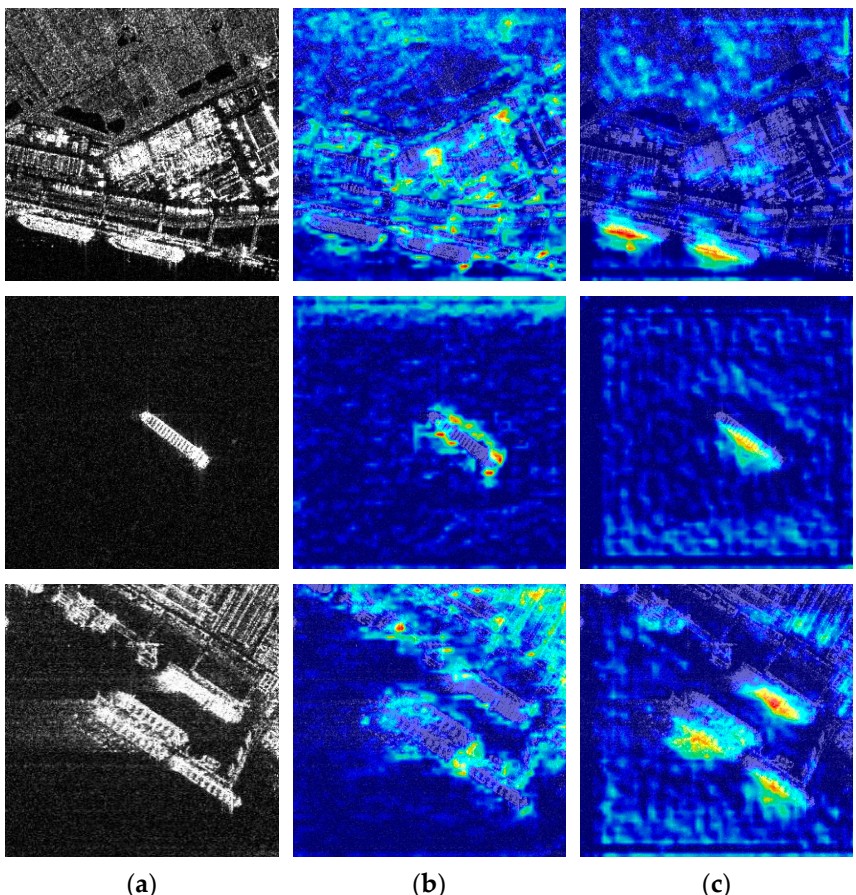

(a)                          (b)                          (c)

**Figure 14.** Visualization of heatmaps for baseline and MT-FANet: (**a**) shows the input SAR image, (**b**) shows the heatmap for the baseline detector, and (**c**) shows the heatmap for MT-FANet.

We also investigated the performance of MT-FANet and the baseline on different backbone networks to explain our choice of backbone network. As shown in Table 8, our

MT-FANet significantly outperformed the baseline regardless of the backbone network which was used. For example, even when using ResNet18, our method showed considerable improvement over the baseline, with an increase in $AP_{50}$ of 5.96% (from 83.25% to 89.21%), demonstrating the significant robustness and effectiveness of our proposed method. Furthermore, we observed that as the feature extraction capability of the backbone network increased, the detection performance of SAR ship targets gradually improved; an example is the increase in $AP_{50}$ from 89.21% to 90.84% and then to 90.72%. Among these networks, ResNet50 achieved comparable performance to ResNet101, with relatively fewer parameters, making it a more efficient choice as a backbone network.

**Table 8.** MT-FANet and baseline ablation study on RSDD-SAR with a different backbone network.

| Backbone | Method | $AP_{50}$ (%) | Recall (%) | F1 | O. $AP_{50}$ (%) | I. $AP_{50}$ (%) | Params (M) | FLOPs (G) |
|---|---|---|---|---|---|---|---|---|
| ResNet101 | Baseline | $85.09 \pm 0.23$ | $88.99 \pm 0.14$ | $86.99 \pm 0.18$ | $91.17 \pm 0.21$ | $58.15 \pm 0.31$ | 49.81 | 71.17 |
| | **Proposed method** | $\mathbf{90.72 \pm 0.24}$ | $\mathbf{91.93 \pm 0.28}$ | $\mathbf{91.31 \pm 0.25}$ | $\mathbf{95.93 \pm 0.23}$ | $\mathbf{67.47 \pm 0.28}$ | **52.72** | **63.43** |
| ResNet50 | Baseline | $84.28 \pm 0.13$ | $88.82 \pm 0.17$ | $86.49 \pm 0.18$ | $90.83 \pm 0.14$ | $55.55 \pm 0.20$ | 30.82 | 51.70 |
| | **Proposed method** | $\mathbf{90.84 \pm 0.18}$ | $\mathbf{92.21 \pm 0.21}$ | $\mathbf{91.52 \pm 0.22}$ | $\mathbf{95.72 \pm 0.19}$ | $\mathbf{66.87 \pm 0.39}$ | **33.73** | **43.96** |
| ResNet18 | Baseline | $83.25 \pm 0.20$ | $87.68 \pm 0.17$ | $85.40 \pm 0.18$ | $90.39 \pm 0.38$ | $51.22 \pm 0.30$ | 17.86 | 38.98 |
| | **Proposed method** | $\mathbf{89.21 \pm 0.14}$ | $\mathbf{90.76 \pm 0.29}$ | $\mathbf{89.98 \pm 0.20}$ | $\mathbf{94.83 \pm 0.12}$ | $\mathbf{61.83 \pm 0.54}$ | **21.12** | **31.60** |

## 5. Discussion

Based on the experimental results, our MT-FANet outperformed other networks in terms of performance metrics while also achieving a better balance between network detection accuracy and efficiency. Our network had the second-fewest parameters and the fewest floating-point operations, yet it was able to achieve the best detection accuracy. Although our MT-FANet achieved good results, as with other current networks, some issues still need to be addressed. Our topology information extraction module used self-attention for construction. Thus, its interpretability needs to be further strengthened. In addition, for densely arranged small targets in SAR images, our model still had some missed detection problems caused by the insufficient feature extraction of small targets. Moreover, our model needs to improve its accuracy in predicting rotation angles, as even minor angle differences can cause significant changes in target accuracy. These problems are important issues that must be addressed in future research. Our research shows that prior information, such as morphology and topology, play an essential role in our network for SAR ship detection. Therefore, future research should focus on using prior knowledge of ship targets to enrich the representations of their features.

## 6. Conclusions

In this article, we propose a novel method called MT-FANet for SAR ship target detection using rotated bounding boxes. Due to the irregular shape of the ship target in SAR images and the fact that the previous ship target detection methods often ignored the inherent topological information of the ship target, the detection performance of the existing methods is not satisfactory. The proposed method uses the morphology features and inherent topological information of ship targets to guide feature extraction and fusion and provides more accurate features for the network to describe the ship target. Specifically, this method first uses ResNet50 as the backbone to extract the original image features, then uses MT-FPN to enhance the morphological features of the ship target, and then deconstructs the morphological features to extract the topological features. The final prediction head (RAFH) completes the detection of ship targets based on the aforementioned morphological and topological features, and uses parameter sharing and two-stage prediction strategies to reduce network complexity and optimize performance. Extensive ablation experiments were conducted to demonstrate the effectiveness of each improvement. The comparative experimental results show that MT-FANet outperformed the other methods on the RSDD-SAR dataset in terms of detection performance.

**Author Contributions:** Conceptualization, Q.L. and D.L.; formal analysis, S.L. (Shuang Liu); funding acquisition, D.L. and S.L. (Suqi Li); methodology, Q.L. and D.L.; project administration, S.L. (Suqi Li); software, R.J.; supervision, H.L.; validation, R.J. and S.L. (Shuang Liu); visualization, R.J. and H.L.; writing—original draft, Q.L. and R.J.; writing—review and editing, S.L. (Shuang Liu). All authors have read and agreed to the published version of the manuscript.

**Funding:** This work was supported by the National Natural Science Foundation of China, grant number 61971075; Basic Scientific Research Project, grant number JCKY2022110C171; Key Laboratory of Cognitive Radio and Information Processing, Ministry of Education, grant number CRKL220202; Sichuan Science and Technology Program, grant number 2022SZYZF02; Opening Project of the Guangxi Wireless Broadband Communication and Signal Processing Key Laboratory, grant number GXKL06200214 and GXKL06200205.

**Data Availability Statement:** No new data were created or analyzed in this study. Data sharing is not applicable to this article.

**Acknowledgments:** The authors would like to thank the anonymous reviewers for their valuable comments.

**Conflicts of Interest:** The authors declare no conflict of interest.

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
