# Peer review of "MT-FANet: A Morphology and Topology-Based Feature Alignment Network for SAR Ship Rotation Detection"

_remotesensing, doi:10.3390/rs15123001_

Round 1

Reviewer 1 Report

 The paper develops MT-FANet which is a new DL architecture that detects ships in the presence of rotations in SAR images, especially for ships with multi-scale and rotational variations in complex backgrounds. It uses deformable convolution for irregularly shaped ship targets and MT-FPN to extract topology information. It then aligns and distinguishes features to predict rotated bounding boxes which improve detection efficiency. Experiments on the RSDD-SAR dataset is offered to demonstrate that the proposed architecture is effective and leads to SOTA performance.
    The paper studies a practically interesting problem and can be followed and can be followed straightforwardly. I have the following comments to be incorporated for the next round of review.     1. One major approach for using deep learning in ship detection problems is using transfer learning methods:   a. Li, Y., Ding, Z., Zhang, C., Wang, Y. and Chen, J., 2019, July. SAR ship detection based on resnet and transfer learning. In IGARSS 2019-2019 IEEE International Geoscience and Remote Sensing Symposium (pp. 1188-1191). IEEE.   b. Rostami, M., Kolouri, S., Eaton, E. and Kim, K., 2019. Deep transfer learning for few-shot SAR image classification. Remote Sensing11(11), p.1374.   c. Lou, X., Liu, Y., Xiong, Z. and Wang, H., 2022. Generative knowledge transfer for ship detection in SAR images. Computers and Electrical Engineering101, p.108041.   d. Lu, C. and Li, W., 2018. Ship classification in high-resolution SAR images via transfer learning with small training dataset. Sensors19(1), p.63.   e. Lang, H., Li, C. and Xu, J., 2022. Multisource heterogeneous transfer learning via feature augmentation for ship classification in SAR imagery. IEEE Transactions on Geoscience and Remote Sensing60, pp.1-14.   I think the above works should be discussed in section 2.1 to give the reader a complete perspective.   2. Could you explain how you tune the hyperparameter lambda in Eq 14?   3. Please run your code several times and report both the average performance and the standard deviation on the tables to allow for making comparisons statistically meaningful.   4. On the tables, I think it is helpful to report F1 score as well.   5. Please release the code on a public domain such as GitHub to make reproducing results easier for other researchers. 

The writing quality is good.

Reviewer 2 Report

SAR ship target detection has always been a challenging task due to the diversity of ship target shapes and serious background interference. The following some revisions should be completed.

1. The abstract part contains some redundant details, such as a rotation alignment feature head (RAFH) and a morphology and topology feature pyramid network (MT-FPN), which do not need to be abbreviated.

2. In Eq. (5), the variable F is unclear. In addition, some of the operation functions and variables in the Figure are not described, and reviewers suggest that authors explain these operations and variables in detail, which is very necessary.

3. In the experiment part, the authors chose ResNet50 as the final backbone network but does not provide corresponding instructions. Please explain in detail.

4. In the method description part, the authors do not provide more details about the network structure, such as the feature map size and channel dimensions of each layer.

5. In Section 4.3, the authors use Average Precision (AP), network parameters (Params), and the number of floating-point operations (FLOPs) as evaluation metrics to compare with other state-of-the-art methods. In order to evaluate the performance of the proposed method more comprehensively, reviewers suggest adding more evaluation metrics, such as recall and F1-score.

SAR ship target detection has always been a challenging task due to the diversity of ship target shapes and serious background interference. The following some revisions should be completed.

1. The abstract part contains some redundant details, such as a rotation alignment feature head (RAFH) and a morphology and topology feature pyramid network (MT-FPN), which do not need to be abbreviated.

2. In Eq. (5), the variable F is unclear. In addition, some of the operation functions and variables in the Figure are not described, and reviewers suggest that authors explain these operations and variables in detail, which is very necessary.

3. In the experiment part, the authors chose ResNet50 as the final backbone network but does not provide corresponding instructions. Please explain in detail.

4. In the method description part, the authors do not provide more details about the network structure, such as the feature map size and channel dimensions of each layer.

5. In Section 4.3, the authors use Average Precision (AP), network parameters (Params), and the number of floating-point operations (FLOPs) as evaluation metrics to compare with other state-of-the-art methods. In order to evaluate the performance of the proposed method more comprehensively, reviewers suggest adding more evaluation metrics, such as recall and F1-score.

Reviewer 3 Report

The paper proposes a feature alignment network for SAR images of ships based on morphology and topology techniques.

The introduction should not contain the discussion about the performances of the current methodologies.

The methodology can be supported by a SAR image example with all steps.

Strong statements are missing the citations.

The Figures must go after the reference in the text.

Figure 10 is missing the units.

Please substitute "our" and "ours" in the tables with "proposed method".

The discussion can be merged with the conclusions.

In general the text can be more concise and fluent using shorter sentences

Reviewer 4 Report

It is somewhat difficult to compare the results in Figures 12 and 13. I think it would be good to compare the results more clearly using colors, scales, lines, etc.

No problems have been found.

Reviewer 5 Report

The paper presents a deep learning network for SAR ship rotation detection, termed as morphology and topology-based feature alignment network (MT-FANet), which utilizes morphological feature and inherent topology structural information. The paper is technically sound, organized, and presented well. 

The following comments have to be considered:

In the abstract section, the following statement should be rewritten to show that the proposed method showed performance in terms of the computed measures, for instance, precision, recall, etc. “The proposed method exhibits superior performance compared to other DL-based algorithms, particularly ship targets that show multi-scale and rotational variations within complex background scenes”.

·      In the introduction section, the paragraphs should be organized into short subparagraphs. For instance, Paragraphs 3, 4 are of crowded which will be confusing for the reader.

·      The related works section is short and lacks to the details on the techniques utilized in the other works. So, this section should be rewritten by adding more details and also a table can be used to present the previously investigated techniques as well as the obtained results.

·      The conclusions section should be also optimized by showing an overview on the findings of the paper.

Minor edits are required.

Round 2

Reviewer 1 Report

The authors have addressed all my concerns.

Reviewer 2 Report

All my concerns are responded. I am satisfactory with the current version.

All my concerns about technical problems are responded. Please revise the English language.